# Targeting Metabolism in Cancer Cells and the Tumour Microenvironment for Cancer Therapy

**DOI:** 10.3390/molecules25204831

**Published:** 2020-10-20

**Authors:** Jiaqi Li, Jie Qing Eu, Li Ren Kong, Lingzhi Wang, Yaw Chyn Lim, Boon Cher Goh, Andrea L. A. Wong

**Affiliations:** 1School of Clinical Medicine, University of Cambridge, Cambridge CB2 0SP, UK; jl2006@cam.ac.uk; 2Cancer Science Institute of Singapore, National University of Singapore, Singapore 117599, Singapore; csiejq@nus.edu.sg (J.Q.E.); csiklr@nus.edu.sg (L.R.K.); csiwl@nus.edu.sg (L.W.); yc.lim@nus.edu.sg (Y.C.L.); phcgbc@nus.edu.sg (B.C.G.); 3Medical Research Council Cancer Unit, University of Cambridge, Cambridge CB2 0XZ, UK; 4Department of Pharmacology, Yong Loo Lin School of Medicine, National University of Singapore, Singapore 117600, Singapore; 5Department of Pathology, National University Health System, Singapore 119074, Singapore; 6Department of Haematology-Oncology, National University Health System, Singapore 119228, Singapore

**Keywords:** cancer cell metabolism, tumour microenvironment, metabolic reprogramming, targeted therapy, immunotherapy

## Abstract

Targeting altered tumour metabolism is an emerging therapeutic strategy for cancer treatment. The metabolic reprogramming that accompanies the development of malignancy creates targetable differences between cancer cells and normal cells, which may be exploited for therapy. There is also emerging evidence regarding the role of stromal components, creating an intricate metabolic network consisting of cancer cells, cancer-associated fibroblasts, endothelial cells, immune cells, and cancer stem cells. This metabolic rewiring and crosstalk with the tumour microenvironment play a key role in cell proliferation, metastasis, and the development of treatment resistance. In this review, we will discuss therapeutic opportunities, which arise from dysregulated metabolism and metabolic crosstalk, highlighting strategies that may aid in the precision targeting of altered tumour metabolism with a focus on combinatorial therapeutic strategies.

## 1. Introduction

To sustain the rapid proliferation characterising cancer cells, corresponding alterations to tumour metabolism must occur to fuel the elevated bioenergetic demands. This understanding has led to the introduction of ‘Deregulating Cellular Energetics’ as a new hallmark of cancer [1]. Initial observations by Otto Warburg described an unusual reliance of cancer cells on glycolysis despite sufficient oxygen, which was later termed the ‘Warburg effect’ to describe this form of aerobic glycolysis [2]. This metabolic reprogramming, while less efficient in terms of ATP production, confers cancer cells with much-needed metabolic intermediates that can be channelled into biosynthetic pathways, such as the pentose-phosphate pathway (PPP) for nucleotide synthesis [3].

While this aerobic ‘Warburg’ glycolytic phenotype had been regarded as the norm in cancer cells, it is becoming increasingly clear that the metabolic needs of tumour cells do not rely on a single metabolic strategy. Recent studies suggest that certain subtypes of cancer cells may preferentially utilize oxidative phosphorylation (OXPHOS) for energy production in glucose-limiting conditions [4]. In addition, OXPHOS dependency may be induced by certain therapies, such as prolonged tyrosine kinase inhibitor (TKI) therapy in certain oncogene-addicted cancers [5,6]. This reflects a phenomenon termed ‘metabolic flexibility’ where cancer cells adjust their metabolic phenotypes in order to gain a selective advantage for cell growth and survival under hostile conditions throughout tumorigenesis up to the time of metastasis [7]. The significance of these metabolic alterations may diverge not only according to intrinsic signalling pathways within the cancer cell, but also rely on the interaction of cancer cells with their surrounding tumour microenvironment (TME), ranging from immune cells and stromal cells to extracellular matrix (ECM) components and soluble factors [8]. There is also emerging evidence to suggest that metabolic reprogramming within cancer stem cell (CSC)-like phenotypes contributes to treatment resistance, therapeutic failure, and cancer relapse.

This review aims to highlight the complex interplay of regulatory cell signalling pathways, interactions between cancer cells and their TME, and the contributions of CSCs to the intricate and coordinated induction of metabolic pathways. Understanding the regulation of cellular metabolism is key for unravelling cancer metabolism as an attractive target for therapeutic exploitation. In particular, we will focus on resistance mechanisms as a result of dysregulated metabolism and metabolic crosstalk, highlighting strategies that may lead to the improved precision targeting of cancer cell metabolism (CCM).

## 2. Altered Cancer Cell Metabolism

### 2.1. Metabolic Dependencies in Cancer Cells

Cancer cells display a distinct metabolic phenotype compared to non-neoplastic cells. As a whole, these changes in metabolic fluxes are achieved by extensive metabolic reprogramming to fuel anabolic growth in nutrient-replete conditions and to support catabolism under nutrient scarcity. Broadly, this involves extracellular uptake of simple nutrients (glucose, amino acids, etc.), which are channelled into biosynthesis via the core metabolic pathways of glycolysis, the tricarboxylic acid (TCA) cycle, the PPP, and non-essential amino acid synthesis, which is followed by subsequent ATP-dependent processes to produce complex biomolecules (Figure 1). Many cancers are known to upregulate glucose consumption, and the classical Warburg phenotype has been reported in a variety of tumour types [9,10,11]. However, a majority of tumours still retain oxidative capacity to produce ATP via OXPHOS [4,12,13,14,15]. Apart from glucose, fatty acids (FAs) and amino acid metabolites are diverted to the TCA cycle to sustain mitochondrial ATP production. Transporters are also upregulated to increase the extracellular uptake of raw materials, including serine and glutamine [16,17,18,19,20,21,22,23]. This not only serves as building blocks for protein synthesis, but also maintains activity of the mTORC signalling system [24]. Subsequently, in a process known as glutaminolysis, glutamine is converted to glutamate, then α-ketoglutarate (α-KG), which serves as another means utilised by cancer cells to fuel the TCA cycle [25] (Figure 1). Glutamine dependency has been reported in non-small cell lung cancer (NSCLC), breast cancer, and brain tumours, and has been associated with greater metastatic potential, therapy resistance, and aggressive clinical phenotype [26,27,28,29]. The specific metabolic dependencies of tumours may be heterogeneous and depend on the driver oncogenes present, microenvironmental interactions, and effect of treatments.

G6P, glucose-6-phosphate; α-KG, α-ketoglutarate; ATP, adenosine 5′-triphosphate; GSH, glutathione; HIF-1, hypoxia-inducible factor-1; mTORC, mTOR complex 1; OXPHOS, oxidative phosphorylation; RTK, receptor tyrosine kinase; SREBP, sterol regulatory element-binding protein.

### 2.2. Metabolic Reprogramming by Oncogenes and Tumour-Suppressor Genes

Despite the great heterogeneity between tumours, metabolic reprogramming seems to involve a common, finite set of pathways to support anabolism, catabolism, and redox homeostasis [30]. Primarily, the PI3K/Akt/mTOR pathway acts as the central regulator of cellular energetics and metabolism, and acts to increase glycolysis and FA synthesis via hypoxia-inducible factor 1-α (HIF-1α) and sterol regulatory element-binding protein (SREBP) activation, respectively [31] (Figure 1). This network is then co-opted by tumours in malignancy, where mutations in the receptor tyrosine kinases upstream of phosphoinositide 3-kinase (PI3K) (e.g., EGFR and HER2), the p110α catalytic subunit of PI3K, the downstream kinase Akt, and the negative PI3K regulator phosphatase and tensin homologue (PTEN) are frequently observed in cancers [32]. Furthermore, many tumours reside in hypoxic environments, as the rapid proliferation exceeds the rate of angiogenesis [33]. To enable successful adaptation to hypoxia, tumours often upregulate HIF-1α signalling, which is a downstream effector of the PI3K/Akt/mTOR pathway.

While HIF-1α activation results in glycolysis, the glucose depletion due to rapid proliferation may lead to reduced energy stores and increased AMP/ATP intracellular levels. This subsequently activates the AMP-activated protein kinase (AMPK)–liver kinase B1 (LKB1) pathway. AMPK/LKB1 activation maintains energy stores by stimulating catabolic pathways that produce ATP, mostly by enhancing OXPHOS and mitochondrial biogenesis. Mechanistically, AMPK enhances sirtuin-1 (SIRT1) activity by increasing cellular NAD^+^ levels, leading to the deacetylation and modulation of the activity of downstream SIRT1 targets especially peroxisome proliferator-activated receptor gamma coactivator 1-α (PGC-1α) and the fork-head transcription factors FOXO1 and FOXO3a, triggering expression of genes that control mitochondrial biogenesis and activity [34].

FA synthesis is regulated by the transcription factor SREBP-1 [35], which regulates enzymes required in the synthesis of FA from acetyl-CoA, enzymes of the PPP for NADPH production, and enzymes that convert acetate and glutamine into acetyl-CoA [36]. Cancers with constitutively elevated rates of FA synthesis utilise mechanisms to keep SREBP-1 active, including sustained mTORC1 signalling, where the effector S6 kinase of mTORC1 activates SREBP-1 and SREBP-2 activity [36]. Elevated PI3K signalling also activates extracellular FA uptake to further sustain intracellular FA levels for FA synthesis [37].

Driver oncogenes are also involved in reprogramming of tumour energetics. Oncogenic *MYC* activity is known to promote aerobic glycolysis through the constitutive elevation of lactate dehydrogenase (LDH) A, upregulation of the glucose transporter GLUT1, and upregulation of several glycolytic enzymes including phosphofructokinase 1 (PFK-1) and enolase [38,39]. MYC has also been implicated in upregulating the uptake and catabolism of glutamine [20]. Specifically, MYC induces expression of genes needed for glutamine metabolism, including glutaminase (*GLS*), glutamine synthetase (*GLUL*), and the glutamine cell-entry transporter *SLC1A5 (ASCT2)* [19,20,40].

Similarly, oncogenic *KRAS* is known to co-opt the metabolic effects of PI3K and MYC pathways to promote tumourigenesis. In *KRAS*-driven pancreatic ductal adenocarcinoma (PDAC), the constitutive MAPK signalling diverts glucose intermediates into hexosamine biosynthesis, and the PPP increases protein glycosylation and nucleotide synthesis [41]. Cells transformed by *KRAS* also show increased expression of genes related to glutamine metabolism and have greater glutamine dependency for anabolic synthesis [42,43]. In addition, the alteration of mitochondrial metabolism by oncogenic *KRAS* promotes carcinogenesis via the activation of growth factor signalling [44]. Finally, tumour-suppressor genes (TSGs) also contribute to the metabolic reprogramming of cancer cells. Loss of p53 triggers OXPHOS [45], and certain tumours are known to retain wild-type p53 to maintain glycolysis, such as in hepatocellular carcinoma (HCC) [46]. Mutant p53 has also been shown to drive Warburg glycolysis [47].

### 2.3. Resistance to Conventional Therapies

Despite advancements in cancer treatment and the availability of multi-modality therapy, development of resistance is still a major barrier contributing to treatment failure. In this section, we will discuss how metabolic reprogramming in cancer cells contributes to therapy resistance.

#### 2.3.1. Resistance to Cell Signalling Pathway Inhibitors

Many cancers demonstrate treatment-induced metabolic adaptation as a mechanism of therapy resistance. In particular, treating oncogene-addicted tumours with TKIs led to resistance development in melanoma and NSCLC, which is accompanied by a metabolic switch to OXPHOS for survival [5,48,49,50,51]. This metabolic switch is thought to contribute to treatment resistance, therapeutic failure, and cancer progression [52]. Treatment of *EGFR*-mutant NSCLC with the 3rd-generation TKI, osimertinib, led to acquired resistance with glycolytic suppression and metabolic switch to OXPHOS [51]. Similar findings have been observed in gefitinib-treated *EGFR*-mutant NSCLC and vemurafenib-treated *BRAF*-mutant melanoma. OXPHOS inhibition restored sensitivity to TKI therapy, and was able to prolong survival and reduce tumour burden in-vivo [6].

Various mechanisms have been proposed to account for the relationship between OXPHOS and TKI resistance. For instance, treatment of *BRAF*-mutant melanomas with BRAF inhibitor, vemurafenib, or with MEK inhibitor, selumetinib, leads to microphthalmia-associated transcription factor (MITF) signalling and elevated expression of the mitochondrial master regulator, PGC-1α. This results in a PGC-1α-mediated induction of an OXPHOS gene programme and mitochondrial genesis [5,49]. Another proposed mechanism of treatment-induced upregulation of OXPHOS is via STAT3 signalling. Various oncogene-addicted cancer cells engage in a positive feedback loop leading to STAT3 activation in response to pathway-targeted therapy, limiting drug response [53]. It has been shown that the non-canonical STAT3 signalling via GRIM-19-dependent import of STAT3 into the mitochondria increases activity of complexes I and II of the electron transport chain (ETC), and, therefore, OXPHOS, leading to TKI therapy resistance [54].

#### 2.3.2. Resistance to Chemotherapy

Metabolic reprogramming in response to conventional chemotherapy have also been described and may potentially be responsible for contributing to the resistant phenotype.

Glucose metabolism contributing to chemotherapy resistance. Enhanced glucose uptake and improved aerobic glycolysis have been shown to contribute to intrinsic and/or acquired resistance to chemotherapy [55,56]. Various glycolytic enzymes have been implicated, including pyruvate kinase muscle isozyme (PKM2). PKM2 catalyses the final step of glycolysis, and, hence, a key regulator of the switch between energy metabolism and anabolic synthesis: either routing glucose metabolism to pyruvate into the TCA cycle, or diverting glucose-derived carbons into other anabolic pathways. PKM2 overexpression has been implicated in carboplatin-resistant NSCLC and is associated with elevated glycolysis compared to parental non-resistant cells [57]. Increased PKM2 is also observed in sera and tissues from colorectal cancer (CRC) patients with poor response to 5-fluorouracil. Thus, PKM2 upregulation may also be linked to 5-fluorouracil resistance in CRC [58]. Other enzymes in glucose metabolism have also been associated with resistance to chemotherapies. Paclitaxel resistance in NSCLC is associated with increased expression of pyruvate dehydrogenase kinase-2 (PDK2) and upregulation of glycolysis. Cisplatin resistance in ovarian cancer was associated with increased expression and activity of glucose-6-phosphate dehydrogenase (G6PD), which enabled greater NADPH production via the PPP for redox homeostasis [55].

Glutamine metabolism contributing to chemotherapy resistance. Elevated glutaminolysis and GSH production is also thought to contribute to chemotherapy resistance. Altered CCM leading to raised GSH levels confers tumours with a greater ability to maintain redox homeostasis [59]. Cisplatin-resistant lung cancer cells have higher levels of glutamate cysteine ligase (GCL), which is the first enzyme of the GSH biosynthetic pathway, and, consequently, elevated GSH. The increased GSH production is thought to counteract the higher levels of reactive oxygen species (ROS) induced by common chemotherapeutic drugs such as cisplatin, and, thereby, protects the cancer cells from oxidative damage [60,61]. Consistent with this, blocking glutamate flux using riluzole was able to selectively kill cisplatin-resistant cells in-vitro and in-vivo [62], and inhibition of GSH biosynthesis with buthionine sulfoximine was found to synergise with cisplatin in breast cancer in-vitro and in-vivo [63].

FA metabolism contributing to chemotherapy resistance. Altered lipid metabolism is another key player in the development of chemoresistance. FA synthase (*FASN*) overexpression induces resistance to anti-tumoral drugs such as doxorubicin and mitoxantrone in breast cancer cells [64], docetaxel resistance in HER2-positive breast cancer [65], gemcitabine resistance in pancreatic cancer [66], and cisplatin resistance in ovarian cancer [67]. *FASN* overexpression is thought to confer tumour cells with an increased survival advantage and reduce apoptosis under the stress of chemotherapy. In breast cancer cells, *FASN* overexpression suppressed drug-induced production of ceramide and, thus, reduced caspase 8-mediated apoptosis under treatment with doxorubicin [64].

## 3. Metabolic Crosstalk with the TME

The homeostasis of the TME is controlled by an intimate crosstalk within and across cancer cells and their various cellular compartments, including endothelial, stromal, and immune cells (Figure 2) [68]. While metabolites that are consumed and released by tumour cells induce changes to TME components in order to support the malignant phenotype, TME cells also play a role in shaping and reprogramming tumour cells by directing paracrine effects, which activate signal transduction.

CAFs, cancer-associated fibroblasts; PDGF, platelet-derived growth factor; TGF-β, transforming growth factor beta; VEGF, vascular endothelial growth factor.

### 3.1. Cancer-Associated Fibroblasts

Often, the rapid growth of solid tumours produces a hypoxic and hypoglycaemic tumour core [69]. While this may be accompanied by aberrant angiogenesis, the vasculature produced are frequently leaky with poor integrity. The resultant hypoxic and nutrient-poor environment hinders tumour growth. Tumour cells overcome this nutrient limitation by reprogramming stromal cells in the TME. Cancer-associated fibroblasts (CAFs) are a key stromal component with a fundamental role in providing metabolic support to tumour cells, thereby facilitating tumour initiation, growth, invasion, and dissemination [70]. This is enabled by metabolic reprogramming of CAFs, releasing energetic substrates into the TME, a phenomenon termed ‘tumour-feeding’ [70,71].

Several modes of tumour-feeding have been postulated (Figure 2). Firstly, in a ‘reverse Warburg effect,’ CAFs undergo metabolic reprogramming switching toward a glycolytic phenotype, whereas the associated cancer cells are reprogrammed toward OXPHOS. Consequently, CAFs produce lactate, which is exported via the monocarboxylate transporter (MCT)-4 into the TME, and taken up by tumour cells via the MCT-1 transporter. Such metabolic coupling have been reported in several tumour types [72,73,74,75]. This is supported in CAFs by an upregulation of glycolysis-related enzymes, such as hexokinase 2 (HK2) and 6-phosphofructokinase liver type (PFKL), and is thought to occur via signalling with platelet-derived growth factor (PDGF) and transforming growth factor beta (TGF-β) [76,77,78].

In addition to lactate, CAFs also supply tumours with glutamine. For example, in regions of glutamine scarcity in the ovarian tumour core, CAFs were reported to undergo metabolic reprogramming to upregulate glutamine anabolism to supply tumour cells [79]. The glutamine released into the TME is subsequently uptaken by cancer cells and converted to glutamate, which fuels the TCA cycle and supports energy production in cancer cells [79].

Apart from the direct release of metabolites into the TME, stromal cells have also been reported to fuel cancer metabolism by releasing metabolites carried in exosomes. These CAF-derived exosomes supply amino acids, lipids, and TCA cycle intermediates to fuel cancer metabolism [80]. CAFs have been reported to release paracrine signals to induce epigenetic changes and metabolic reprogramming of PDAC cells, resulting in changes similar to *KRAS*-driven oncogenic transformations by increasing anabolic metabolism and pro-tumourigenic changes in cancer cells [81].

### 3.2. Endothelial Cells

Tumour angiogenesis is the proliferation of a network of blood vessels that provide oxygen and nutrient support for tumours. In the past, the angiogenic switch was thought to be mediated by angiogenic molecules. However, it is now evident that distinct metabolic signatures of endothelial cells (ECs) are deregulated in cancer and vascular EC function can be modulated by metabolites [82] (Figure 2).

ECs are largely glycolytic and depend on glucose for proliferation. This phenotype is promoted by tumour paracrine signalling. For example, conditioned media from hypoxic glioma cells induced ECs to upregulate surface GLUT1 to enhance glucose uptake [83]. In addition, vascular endothelial growth factor (VEGF) signalling by tumours causes 6-phosphofructo-2-kinase/fructose-2,6-biphosphatase 3 *(PFKFB3)* upregulation in ECs, which activates PFK-1 to further augment the glycolytic phenotype [84]. Lactate is also enriched in the TME, which can trigger tube formation in ECs via HIF-1α-dependent NF-κB activation [85,86].

Tumours also modulate amino acid availability in the TME, which have pleiotropic effects in vessel sprouting. ECs require glutamine for TCA cycle anaplerosis and non-essential amino acid synthesis. Depriving ECs of glutamine or inhibition of *GLS1*, thus, causes vessel sprouting defects due to impaired EC proliferation and migration [87,88]. VEGF signalling also requires glycine to promote angiogenesis and serine is also required for mitochondrial function in ECs [89,90]. Thus, tumour-dependent depletion of amino acids in the TME contributes to vascular defects and aberrant angiogenesis seen in some tumours.

Finally, FAs supply the carbon needed for dNTP synthesis during EC sprouting [91]. FA carbons are used by ECs to replenish the TCA cycle, and are incorporated into aspartate (nucleotide precursor) and uridine monophosphate (precursor in pyrimidine synthesis), supplying adequate dNTPs for proliferation [91]. Altered CCM may, therefore, alter FA availability to support EC proliferation.

## 4. Tumour Immune Microenvironment

The immune system interacts intimately with tumour development in a complex, bidirectional crosstalk that can both inhibit and enhance tumour growth and progression. This interaction has gained recognition as a hallmark of cancer and immunotherapy has become an established pillar of cancer therapy [1]. Immune cells execute their function most effectively when they are able to respond swiftly to environmental stimuli through phenotypic shifts, enhanced by the radical reprogramming of immune cell metabolism [92]. On the other hand, impaired metabolic flexibility results in an ineffective anti-tumour immune response, and may be explained by the mutual metabolic requirements of immune cells and tumour cells, which compete for similar essential nutrients such as glucose and glutamine. Besides nutrient availability, high production of metabolites such as lactate, kynurenine, and other metabolic by-products of cancer metabolism can be harmful for immune cells, resulting in tumour immunosuppression [93].

### 4.1. T Cells

The effects of altered tumour metabolism on T cell function is summarised in Figure 3.

A2AR: Adenosine 2A receptor; AHR: aryl hydrocarbon receptor; Csk: C-terminal Src kinase; IDO: indoleamine-pyrrole 2,3-dioxygenase; Lck: lymphocyte-specific protein tyrosine kinase; mTORC1: mammalian target of rapamycin complex 1; NFAT: nuclear factor of activated T-cells; OXPHOS: oxidative phosphorylation; TCA: tricarboxylic acid cycle; Teff; effector T cells; Treg: regulatory T cells.

#### 4.1.1. Altered CCM Deprives T cells of Nutrients Essential for Anti-Tumour Activity and Induces Polarisation of Immunosuppressive T Cell Subsets

When a T cell is activated, there is a dramatic metabolic reprogramming mediated through the PI3K/Akt/mTOR pathway, greatly reminiscent of the metabolic reprogramming observed in cancer cells. This ‘Warburg phenotype’ adopted by activated T cells involves upregulation of aerobic glycolysis, increased glucose metabolism through the PPP, increased glutaminolysis, and increased FA synthesis. This leads to a competition between effector T cells (Teff) and tumour cells for similar nutrients especially glucose, thereby impairing the Teff anti-tumour response. Furthermore, glucose limitation is reported to produce an ‘exhausted’ T cell phenotype, characterised by increased programmed cell death protein 1 (PD-1) expression on the surfaces of T cells, which accounts for the greater proportion of ‘exhausted’ T cells in tumours and leads to cancer immune evasion [94,95,96]. Tumour-mediated T cell exhaustion is also characterised by the upregulation of other inhibitory receptors including cytotoxic T lymphocyte-associated antigen 4 (CTLA-4), T-cell immunoglobulin, and mucin-domain containing-3 (TIM-3), lymphocyte activation gene-3 (LAG-3), and T cell immunoreceptor with Ig and immunoreceptor tyrosine-based inhibitory motif (ITIM) domains (TIGIT) [97,98,99,100]. Decreased glucose metabolism was also found to impair the epigenetic reprogramming required for T cell activation. Reduced flux through the glycolytic pathway leads to insufficient acetyl-CoA to maintain α-KG levels required for cofactor function for histone acetylation, resulting in reduced interferon γ (IFNγ) expression and impaired helper T cell (Th1) activity [101].

Similar to glucose, activated T cells have a higher requirement for amino acids, most notably glutamine. Glutamine is utilised by active T cells and is required for the inflammatory responses of Th1 and Th17 cells [102,103], and decreased glutamine availability in the TME is reported to blunt anti-tumour immunity by limiting essential biosynthetic pathways for T cell proliferation. A concern with targeting metabolic pathways is the extensive overlap between the metabolic phenotypes of tumour cells and activated immune cells. Theoretically, GLS inhibition can limit T cell metabolism along with crippling tumour metabolism since increased glutaminolysis is a hallmark of both tumour cells and activated T cells. However, Leone et al., 2019, showed that, while glutamine blockade in cancer cells led to suppression of oxidative and glycolytic metabolism, by contrast CD8^+^ T cells responded by upregulating acetate metabolism, generating high levels of acetyl-CoA for direct fuelling of the TCA cycle as well as indirect fuelling via increased glucose anaplerosis through pyruvate carboxylase activation. These resulted in upregulation of oxidative metabolism with CD8^+^ T cells adopting a long-lived, highly activated phenotype [104]. These divergent responses to GLS inhibition serves as a ‘metabolic checkpoint’ and an opportunity to simultaneously inhibit tumour metabolism while boosting anti-tumour immune activity. Other amino acids required for T cell activity includes arginine and tryptophan. Tumour-depletion of arginine in the TME can impair T cell anti-tumour immunity, particularly memory T cell immunity [105]. Tryptophan deficiency is also known to inhibit mTORC1 activity in T cells, impairing T cell activation and proliferation [106].

The effect of FAs is less well-characterised because different T cell subsets utilise FAs differently. Of note, memory T cells are more dependent on FA oxidation (FAO) for energy and are unable to develop in the absence of FAs in culture. However, recent studies suggest that FAs used by memory T cells for FAO are derived from extracellular glucose, rather than direct utilisation of extracellular FAs [107]. Currently, the role of FAs in T cell metabolism is unclear, and further studies are required.

In addition, nutrient depletion in the TME alters T cell differentiation and induces the polarisation of immunosuppressive T cell subsets [108,109]. Glucose deficiency enriches for regulatory T cells (Tregs) because, in contrast to Teffs that rely on aerobic glycolysis, Tregs rely more on FAO. FOXP3 metabolic reprogramming leads to MYC and glycolysis suppression, which enhances OXPHOS and NADH oxidation. These adaptions confer Tregs a metabolic advantage in the low-glucose, high-lactate microenvironment in the TME, shifting the balance to favour Treg enrichment over Teffs and facilitating tumour immune evasion [110]. Glutamine deficiency in the TME may also shift T cell differentiation toward a pro-tumourigenic phenotype, as glutamine deficiency disproportionately impairs Th1 and Th17 subsets more than Tregs, thereby enriching for Tregs in the TME [108].

#### 4.1.2. CCM-Derived ‘Waste’ Metabolites Inhibit T Cell Function and Promotes T Cell Exhaustion

Lactate is reported to inhibit T cell proliferation and cytokine production [111,112]. Tumour-derived lactate accumulates in the TME, leading to impaired T cell export of lactate and intracellular build-up. Elevated lactate suppresses glycolytic enzymes via end-product inhibition, impairing T cell metabolism and function. Lactate build-up in the TME also causes T cell acidification, preventing translocation of nuclear factor of activated T cells (NFAT) into the nucleus and NFAT-mediated transcription. This, thus, inhibits IFNγ production and impairs T cell response [112]. Finally, lactate is reported to inhibit the PI3K/Akt/mTOR pathway in T cells, blunting T cell activation [113,114].

Other tumour-derived ‘waste’ metabolites are also suggested to play a key role in T cell immunosuppression (Figure 3). Adenosine is released by tumours into the TME and inhibits the T cell anti-tumour response. Upon binding to Adenosine A2A receptor (A2AR), signalling leads to an increase in cAMP levels, protein kinase A (PKA) phosphorylation of Csk, which subsequently inhibits Lck and antagonising TCR signalling. This leads to reduced T cell activation, cytokine production, and anti-tumour immunity [115]. Kynurenine, the first breakdown product in indoleamine 2,3-dioxygenase (IDO)-dependent tryptophan degradation, has also been reported to exert immunosuppressive effects and induce T cell apoptosis [106,116].

### 4.2. Myeloid-Derived Suppressor Cells, Tumour-Associated Macrophages, and Dendritic Cells

Myeloid-derived suppressor cells (MDSCs) are a heterogeneous population of cells of myeloid origin that contribute to TME immunosuppression and exert suppression on T cell and innate immune cell responses. The altered CCM environment influences MDSC functionality, which can further bolster their immunosuppressive effects.

For example, the hypoxic TME leads to HIF-1α signalling, which aids in MDSC differentiation to tumour-promoting tumour-associated macrophages (TAMs) [117]. Lactate similarly induces polarisation toward the pro-tumourigenic M2 macrophage phenotype via HIF-1α signalling [118], and induces upregulation of PD-L1 on myeloid cells, facilitating Teff suppression [119]. Furthermore, hypoxia and lactate in the TME induce a metabolic switch from glycolysis toward OXPHOS, which is consistent with the enrichment and continued functionality of MDSCs and TAMs in a primarily hypoglycaemic TME [120].

The anti-tumour functions of macrophages are also inhibited by altered cancer metabolism. Extracellular lactate reduces activation of monocytes, as measured by reduced glycolysis-dependent tumour necrosis factor (TNF) production [121]. Tumour prostaglandin E_2_ (PGE_2_) production is also found to subvert myeloid cell function. This was reported in various oncogene-addicted tumour models [122]. For example, in a *BRAF*-mutant model of melanoma, and *NRAS*-mutant models of melanoma, breast, and CRC, PGE_2_ production impairs myeloid cell activation especially the antigen-presenting ability required for T cell activation [122].

Dendritic cells (DCs) are also key players in anti-tumour immunity. Activation of DCs involves metabolic reprogramming not unlike that of T cells, switching from OXPHOS to aerobic glycolysis. Competition with tumour cells in the TME for essential nutrients, in particular glucose, can severely limit DC activity and antigen-presenting ability [123]. In addition, a low energy state leads to elevated AMPK signalling, which inhibits glycolysis and promotes greater OXPHOS and FAO. This is reminiscent of a tolerogenic DC phenotype [124]. In addition, FAO induction in tumour-associated DCs (TADCs) was found to drive the production of IDO, which results in Treg polarisation and further immunosuppression of T cells in a model of melanoma [125]. Expression of the inhibitory receptor CTLA-4 on Tregs can also induce IDO activity by DCs [126]. This immunosuppressive crosstalk between dysregulated immune cells serve to drive a positive-feedback loop whereby immunosuppression is self-maintained and further propagated in the TME [123].

### 4.3. Natural Killer Cells and Neutrophils

Other cells of the innate immune system are also intricately linked to metabolic changes in the TME. Due to the greater energetic demands of natural killer (NK) cells, in particular increased glycolysis, NK cells are also subject to competition with tumour cells for glucose. Thus, the perennial problem of glucose and nutrient deprivation in the TME also impairs NK function. Furthermore, the aberrant production of metabolites as a consequence of altered CCM also impacts NK activity. Metabolic reprogramming of NK cells upon activation requires the SREBP transcription factors. 25-hydroxycholesterol (25-HC) is a cholesterol-derived metabolite produced by various cancers, such as glioblastoma [127], and can inhibit translocation of SREBP from the ER to the Golgi, impairing NK activation [128]. Elevated lactate in the TME also reduces NFAT signalling in NK cells, reducing IFNγ production, CD25 levels, and tumour-killing capabilities [112].

Neutrophils are frequently discounted from a metabolic perspective as purely glycolytic. The low glucose availability in the TME is predicted to limit neutrophil ROS production, which can disrupt CD4+ T cell viability and function. However, tumour-directed metabolic reprogramming can switch neutrophils to an oxidative phenotype. For instance, 4T1 tumours was found to induce a metabolic shift to produce mitochondria-rich oxidative neutrophils through aberrant stem cell factor (SCF)/c-Kit signalling. Oxidative neutrophils can use mitochondrial FAO to support NADPH oxidase-dependent ROS production in the hypoglycaemic TME. Thus, tumour-mediated SCF/c-Kit signalling can induce an oxidative phenotype in neutrophils to overcome to metabolic limitations, resulting in maintained ROS production despite the hypoglycaemic TME, and, hence, sustained immunosuppression [129].

### 4.4. PD-1 and CTLA-4 Signalling and the Effects of Immune Checkpoint Blockade on Metabolic Pathways

PD-1 and CTLA-4 are immune checkpoints that serve as negative regulators of T cell function [130]. Signalling via PD-1 limits T cell activation, preventing excess inflammation and tissue damage [131]. This regulatory function is hijacked by cancer cells that upregulate the ligands PD-L1 and PD-L2 on their surface to dampen anti-tumour immunity [132]. PD-1 contains two intracellular tyrosine motifs that, when engaged by its ligands, result in phosphorylation of tyrosine residues, leading to recruitment of protein tyrosine phosphatases (PTPs) such as SHP2 [132]. PTPs antagonise positive signals from the TCR and CD28, hence antagonising downstream pathways including PI3K/Akt, Ras, ERK, Vav, and PLCγ, which are key pathways required for metabolic reprogramming of activated T cells [133,134,135].

Cancer therapy has entered the ‘immunotherapy era’ with anti-PD-1/PD-L1 and anti-CTLA-4 antibodies being incorporated into the treatment for melanoma, triple-negative breast cancer (TNBC), NSCLC, and metastatic renal cell carcinoma (RCC), among others. Such immune checkpoint blockade (ICB) therapy, aimed at reversing the immune suppression caused by tumour cells, also shapes the TME by affecting tumour metabolism [136]. Ligation of PD-1 on activated T cells impair glycolysis or amino acid metabolism [137], while PD-1 blockade upregulates *GLUT1* to restore glucose uptake, promoting glycolysis in effector T cells [94]. Furthermore, PD-1 promotes FAO of endogenous lipids by increasing expression of *CTP1A* and upregulating lipolysis [137]. ROS generation by activators of mTOR, AMPK, and PGC-1α were found to synergise with PD-1 blockade [138]. Taken together, this strengthens the role of combining PD-1 blocking therapies with metabolism-based therapies for more efficacious anti-tumour immunity.

Ligation of CTLA-4 also leads to similar inhibition of key metabolic reprogramming. However, CTLA-4 signalling on T cells inhibits glycolysis without augmenting FAO [137]. This suggests diverging roles of these two immune checkpoints: CTLA-4 sustains the metabolic profile of non-activated cells, while PD-1 functions to dampen metabolic reprogramming in activated cells [137]. Regardless, the function of PD-1 and CTLA-4 in antagonising key metabolic pathways in T cells are mechanisms tumours used to limit anti-tumour immunity, and provides an explanation for the capacity of T cells to be metabolically invigorated by ICB.

### 4.5. Resistance to Immunotherapies

However, in reality, only a small proportion of patients respond well to ICB [139]. Studies have uncovered several reasons for this gap, including poor tumour immunogenicity, tumour editing, and lack of sufficient tumour-infiltrating T cells in a ‘cold’ tumour [140,141].

The altered metabolism in cancer cells is often associated with dysregulated expression of key metabolic enzymes. The aberrantly expressed enzymes have pleiotropic effects that contribute to immunosuppression, limiting the effectiveness of immunotherapies. For example, many cancers display *MYC*-dependent upregulation of the alternatively spliced PKM2 enzyme as a mechanism to enhance aerobic glycolysis [142]. Furthermore, independent of its enzymatic action on glycolysis, PKM2 promotes the expression of PD-L1 on tumour surfaces and, hence, promotes immune suppression [143]. PKM2 activity also aids in recruitment of MDSCs, and is associated with increased metastasis and poor prognosis in HCC [144].

Tumour metabolism also limits the effectiveness of immunotherapies by affecting the tumour mutation rate and antigenicity [145]. Metabolism is tightly linked to DNA repair through chromatin remodelling, epigenetic modifications, and regulation of the redox status [146]. Altered tumour metabolism can promote chromatin remodelling and epigenetic modifications in multiple ways, such as by supplying acetyl and methyl groups and producing metabolites that act as key cofactors or inhibitors of epigenetic enzymes, such as α-KG, succinate, fumarate, and 2-hydroxyglutarate [147]. Furthermore, the enhanced nucleotide biosynthesis in tumours promotes DNA repair [92]. Taken together, these processes lead to a reduced mutation rate and, hence, reduced tumour antigenicity, thereby limiting the effectiveness of immunotherapies.

Finally, CAFs in the TME may also contribute to immunotherapy resistance by several mechanisms. Firstly, the release of immunosuppressive cytokines TGF-β and IL-6 by CAFs lead to reduced proliferation and trafficking capacity of antigen-presenting DCs, thereby impairing T cell priming against tumour antigens [145,148]. CAFs also directly upregulate immune checkpoint ligands on their surface, including PD-L1 and PD-L2 [149,150]. Next, CAFs impair T cell migration to the tumour bed. Through tight regulation of the local chemokine gradient, CAFs limit T cell attraction to the TME [151,152]. CAFs also impair T cell access to the tumour, directly via inhibition of T cell migration through a TGF-β-dependent gene programme [153] as well as indirectly by altering the composition of the ECM, creating a denser ECM network, which functions as a physical barrier to T cell infiltration [154,155].

## 5. Cancer Stem Cells and Metabolic Reprogramming in CSCs as a Mechanism of Therapeutic Resistance

The CSC model postulates that a small, metabolically distinct, and quiescent CSC population is responsible for resistance to therapies that target rapid proliferation [156]. This adds further to the complexity of the TME as CSCs reside in, actively remodel, and are reciprocally modulated by each element of the TME. This leads to an intricate crosstalk between CSCs and cancer cells, stromal components, and the immune milieu, generating a wide variety of resistance mechanisms [157].

A number of studies suggest stronger glycolytic metabolism in CSCs as compared to differentiated tumour cells [158,159,160]. Interestingly, elevated MYC was independently identified as the main driver of stemness in all of these cancer types [159]. This has been linked to a MYC-driven glycolytic programme, consistent with what is observed in induced pluripotent stem cells [160], leading to the notion that, rather than distinct metabolism being a ‘by-product’ of cancer stemness, metabolism may be the ‘driver’ controlling stemness characteristics. On the other hand, emerging evidence suggests that CSCs demonstrate extensive metabolic flexibility and acquiring increased oxidative metabolism confers greater ability to overcome therapy-induced stressful metabolic environments promoting CSC survival.

Therapy-induced enrichment of CSC populations was observed in various cancer models [6]. In a model of BCR-ABL driven chronic myeloid leukaemia (CML), persistent leukaemic stem cells (LSC) responsible for treatment resistance and relapse depended on OXPHOS upregulation for survival. Subsequently, combination therapy of OXPHOS inhibition plus BCR-ABL targeted therapy was able to selectively eradicate CML LSCs in-vitro and in-vivo [161]. Similarly, in a mouse model of PDAC, resistant cells surviving *KRAS* ablation showed more features of CSCs and reliance on OXPHOS for survival. Treatment with OXPHOS inhibitors also resulted in eradication of CSCs in-vitro and in-vivo [162]. Mechanistically, the metabolic switch to OXPHOS in pancreatic CSCs was found to be dependent on the MYC/PGC-1α balance [163]. Suppression of MYC and subsequent increase in PGC-1α stimulated PGC-1α-dependent mitochondrial biogenesis and OXPHOS dependency. Consequently, treatment of pancreatic CSCs with metformin led to an energy crisis and apoptosis [163].

## 6. Therapeutic Opportunities Targeting Altered CCM

With greater understanding of dysregulated cancer metabolism and the metabolic interplay of cancer cells, TME and CSCs, therapies can be developed to target these processes and overcome therapeutic resistance. In the following sections, we will highlight promising strategies that target altered pathways of metabolism, alone or in combination with other available anti-cancer therapies.

### 6.1. Glycolysis Inhibitors

Hexokinase catalyses the first step in glycolysis of which the HK2 isoform is upregulated by many tumours and is needed to maintain the high glycolytic rate. HK2 is, thus, a potential target for inhibition [164]. Various HK2 inhibitors have been identified, including 2-deoxyglucose (2-DG), 3-bromopyruvate (3-BP), and lonidamine (LND). In particular, 2-DG is a glucose mimetic that competitively inhibits the production of glucose-6-phosphate (G6P) from glucose and causes ATP depletion and cell death [165]. As a single-agent therapy, 2-DG had shown promising results and reached phase I/II clinical trials for the treatment of solid tumours and hormone refractory prostate cancer, but, unfortunately, was halted due to limited efficacy on tumour growth and significant toxicities (NCT00633087). Detailed phase II and III clinical trials have also been performed for LND in several tumour types. Unfortunately, LND only showed modest clinical activity, and further research was halted due to concerns over liver enzyme abnormalities and its lack of specificity [166].

PFKFB3 is a potent regulator of glycolysis and is frequently upregulated in cancers [167]. Various inhibitors of PFKFB3 have been reported, including the weak PFKFB3 inhibitor, 3-(3-pyridinyl)-1-(4-pyridinyl)-2-propen-1-one (3PO). A derivative of 3PO, PFK15, showed improved pharmacokinetic and anti-neoplastic properties in-vitro and in-vivo. PFK15 was able to cause a rapid induction of apoptosis in transformed cells and showed anti-tumour effects in-vivo [168]. Another 3PO derivative, PFK158, is currently under evaluation in a phase I trials for advanced solid malignancies (NCT02044861) [169].

Finally, small molecular inhibitors of GLUT transporters are undergoing evaluation. Phloretin antagonises GLUT2 in TNBC and suppressed cell growth and metastasis of TNBC in-vitro and in-vivo [170]. STF-31 is a selective GLUT1 inhibitor, which showed effectiveness in Von Hippel–Lindau-dependent RCC models [171]. Another GLUT1 inhibitor, WZB117, was able to inhibit cancer growth and viability in-vivo, and was synergistic with cisplatin and paclitaxel [172]. GLUT3 inhibition has also been shown to be effective in delaying the resistance to temozolamide in the treatment of glioblastoma multiforme (GBM) [173]. The FDA-approved antiviral drug, ritonavir, was found to have GLUT4 inhibitory activity in multiple myeloma (MM) and is synergistic with metformin [174,175]. Of note, tumours expressing high levels of the cystine-glutamate antiporter xCT (SLC7A11) are heavily dependent on the PPP to supply reducing power in the form of NADPH [176], and GLUT inhibition selectively kills SLC7A11 high cancer cells in-vitro and in-vivo [177], presenting a metabolic vulnerability that can be targeted. Finally, an ongoing phase I trial is evaluating the role of ritonavir in combination with metformin in treating patients with relapsed or refractory MM or chronic lymphocytic leukaemia (CLL) (NCT02948283). Further work on small molecule GLUT inhibitors are required to establish safety and efficacy for translation to clinic.

### 6.2. OXPHOS Inhibitors

Strategies for inhibiting OXPHOS range from direct or indirect inhibition of mitochondrial respiratory chain complexes or inhibiting mitochondrial biosynthesis.

#### 6.2.1. Biguanides

Of the complex I inhibitors, the biguanides, metformin, and phenformin, have been most extensively studied in cancer prevention and treatment [178,179]. Their anti-tumour activity relates to mTOR inhibition by the activation of AMPK and LKB1, thus reducing cellular proliferation. Proof-of-concept studies have confirmed biological evidence supporting the anti-proliferative effects of metformin in endometrial cancers, which are strongly associated with aberrations in the PI3K/AKT/mTOR pathway [180]. Observational studies have reported that the likelihood of developing cancer in type II diabetic patients treated with metformin is 30% lower than that of patients taking alternative oral hypoglycaemic agents (OHGAs). In a meta-analysis, metformin was found to be a useful adjuvant agent in preventing cancer relapse with the greatest benefits seen in prostate cancer and CRC [181]. Multiple trials investigating the effect of metformin as monotherapy and in combination with chemotherapy or TKI therapy are underway (e.g., NCT03137186, NCT01243385). However, there are doubts on whether metformin is able to reach sufficiently high concentrations to inhibit OXPHOS in-vivo. This is due to the requirement of uptake via organic cation transporters (OCTs), which may be reduced in various tumour types [182]. Phenformin, on the other hand, may have intrinsic pharmacokinetic properties to overcome this limitation. Being more hydrophobic, phenformin is able to cross biological membranes without requiring active transport. It is, therefore, nearly 50 times as potent as metformin due to its higher absorption and tissue bioavailability [183]. Phenformin had demonstrated more potent inhibition of cell proliferation compared to metformin in multiple tumour types (breast, lung, colon, melanoma, GBM, and prostate) [183]. Phenformin is also proposed to delay treatment resistance to conventional cancer therapies. For instance, in osimertinib-resistant EGFR-mutant NSCLC, which is reported to undergo a metabolic switch from glycolysis to OXPHOS, phenformin was able to delay the long-term development of osimertinib resistance [51]. There are also proposed biomarkers to indicate susceptibility and increase the efficacy of biguanide therapy. For example, LKB1 deficient cells show increased sensitivity to metabolic stress and were predictive of susceptibility to phenformin therapy, supporting its role as a potential biomarker for OXPHOS inhibition [179]. Unfortunately, due to its association with a higher incidence of lactic acidosis, phenformin had been withdrawn from clinical use as an OHGA and, hence, limited clinical studies are available looking into the effect of phenformin for cancer therapy [184].

#### 6.2.2. IACS and Other Complex I Inhibitors

Another complex I inhibitor, IACS-010759, recently demonstrated preclinical efficacy in inhibiting growth of CLL and acute myeloid leukaemia (AML) [185]. IACS-010759 is potent and can be administered orally, and has progressed to phase I trials in advanced solid tumours, which was reported to be well-tolerated with preliminary evidence of anti-tumour activity. Maximum tolerated dose (MTD) expansions are planned for patients with TNBC, pancreatic cancer, and castration-resistant prostate cancer. Given the diverse metabolic dependencies of tumours, it is crucial to have means to stratify tumours to better predict their susceptibility to metabolism-based treatments. The loss of enolase 1 (*ENO1)* was found to be predictive of sensitivity to IACS-010759 in some brain tumour cell lines (D423, Gli56) [186]. Loss of *SMARCA4*, a component of the SWI/SNF chromatin remodelling complex, also results in greater reliance on OXPHOS, and cells are more sensitive to OXPHOS inhibition by IACS-010759 in NSCLC [187]. These findings have led to plans to investigate the efficacy of IACS-010759 in molecularly-selected tumours (*ENO1* loss and *SMARCA4* mutation) (NCT03291938, NCT02882321) [188].

Some of the other complex I inhibitors initially demonstrated promising results but were subsequently withdrawn due to toxicity. For instance, BAY87-2243 had advanced to phase I studies (NCT01297530). However, the trial was terminated due to significant toxicities (grade III nausea/vomiting). A phase Ib study was completed for ME-344. However, further research halted due to significant grade III/IV toxicities and lack of clinical efficacy in unselected patients with small cell lung cancer, ovarian cancer, and cervical cancers [189]. Other complex I inhibitors include carboxyamidotriazole (CAI). In particular, CAI had completed phase III clinical trial for advanced NSCLC (NCT00003869). Unfortunately, no additional clinical benefit was reported with the addition of CAI over placebo following chemotherapy [190].

#### 6.2.3. Complex II-V Inhibitors

Apart from complex I inhibitors, various compounds that inhibit complexes II-V of the mitochondrial ETC have also been investigated. These include the low-affinity complex II inhibitor α-tocopheryl succinate, complex III inhibitor atovaquone, and the complex IV inhibitor arsenic trioxide, which is used for treating acute promyelocytic leukaemia [4]. However, many of these compounds have pleiotropic effects apart from mitochondrial inhibition, and it may be difficult to distil and attribute their therapeutic effects to OXPHOS inhibition. Presently, limited data is available studying the anti-tumour efficacy of complex II-IV inhibitors, and further research is warranted.

A novel Complex V (F_0_F_1_ ATP synthase) inhibitor, gboxin, selectively accumulates in the mitochondria due to its positive charge. This leads to increased proton gradient and pH in the cancer cell mitochondria, thereby inhibiting ATP production via the ATP synthase [191]. In GBM cells, gboxin rapidly and irreversibly compromises oxygen consumption, leading to gboxin-mediated cell death [191].

#### 6.2.4. Indirect Inhibition of Mitochondrial Complexes and Mitochondrial Protein Synthesis

Treatment-induced metabolic switch to OXPHOS has been hypothesized to involve mitochondrial STAT3 (mSTAT3), which indirectly promotes OXPHOS by interacting with retinoic-interferon-induced mortality 19 (GRIM-19) and the ETC complexes I and II [192]. Thus, indirect OXPHOS inhibition can also be achieved by mSTAT3 inhibition. OPB compounds able to indirectly inhibit OXPHOS via their action on mSTAT3, with several OPB compounds reaching clinical trials (OPB-51602, OPB-111077, OPB-31121) in treatment-refractory solid tumours as well as in haematological malignancies. OPB-51602 had been evaluated in a phase I first-in-human study (NCT01184807), demonstrating promising anti-tumour activity in *EGFR*-mutant NSCLC with prior EGFR-TKI exposure [193]. OPB-111077, a second-generation compound with an improved safety profile, had completed phase I evaluation in treatment-refractory solid tumours [194]. Further trials are underway in patients with diffuse large B-cell and oncogene-addicted solid tumours (NCT03158324) [194].

Mitochondrial dysfunction may also be induced by inhibiting mitochondrial protein translation. This can be achieved by using the antibiotic tigecycline, which was identified in a screen with OXPHOS-dependent leukaemia cells [195]. Subsequent studies using tigecycline to treat TKI-resistant CML cells were able to successfully eliminate the OXPHOS-dependent CSC population thought to be responsible for treatment resistance [161]. Another antibiotic, salinomycin, inhibits OXPHOS and was able to eliminate the CSC gene expression signature in in-vivo studies of breast cancer [196]. Mitochondrial metabolism can also be targeted by the mitochondrial chaperone TRAP1 inhibitor, gamitrinib, which leads to impairments to mitochondrial protein folding [197].

### 6.3. Glutamine Blockade

The pleiotropic effects of glutamine metabolism on cancer proliferation and signalling makes glutamine blockade another potential strategy for targeting cancer metabolism. Proposed strategies include depleting cancer cell glutamine supply, blocking glutamine uptake transporters, using glutamine mimetics as anti-metabolites, and the most promising, selective inhibition of GLS.

The mitochondrial enzyme GLS is a key component of glutaminolysis, which produces α-KG for the replenishment of TCA cycle intermediates. Small molecule GLS inhibitors such as bis-2-(5-phenylacetamido-1,2,4-thiadiazol-2-yl)ethyl sulfide (BPTES), CB-839, and compound 968, are able to inhibit GLS isoforms not commonly expressed in normal cells, allowing for greater selectivity in targeting cancer cells while reducing toxicity to normal cells [198,199]. The GLS1 inhibitor, CB-839, has reached the furthest in the developmental pipeline, and is selective, more potent, and demonstrates greater bioavailability compared to BPTES [200,201]. In early phase studies, CB-839 showed safety and tolerability in solid tumours with promising signs of clinical activity in multiple tumour types including TNBC, NSCLC, and mesothelioma (NCT02071862) [202,203]. In particular, for RCC, radiographic stable disease (SD) or partial response (PR) was observed in 9 of 15 (60%) efficacy-evaluable RCC patients [202]. No specific biomarker of patient selection for GLS inhibition has been established, but studies have evaluated tumour *GLS* overexpression and the specific GLS1 variant that is overexpressed. GLS1 exists two main splice variants: KGA, the full length GLS1, and GAC, which has an alternative carboxy-terminus and a lower molecular weight [204]. Of these, only the GAC splice variant is sensitive to GLS inhibition by CB-839 [200].

Depletion of the cancer cell glutamine supply can be achieved via L-asparaginase, a recognised treatment for acute lymphoblastic leukaemia (ALL), which removes the amide nitrogen from glutamine to form glutamic acid [205]. Serum L-asparaginase activity strongly correlates with glutamine depletion in the blood and improved treatment outcomes in ALL patients [206].

Inhibitors have been developed for the glutamine uptake transporter SLC1A5 (ASCT2), which is upregulated across many tumour types to increase glutamine uptake [207,208]. A pharmacological agent used as a common preclinical tool to inhibit SLC1A5 is the L-glutamine analogue, l-γ-glutamyl-p-nitroanilide (GPNA). Pharmacological inhibition of SLC1A5 with GPNA was able to decrease lung cancer cell growth and viability by inhibiting glutamine-dependent mTORC1 signalling [209,210]. In an immunohistochemistry study on NSCLC patients, SLC1A5 protein expression was found to be a significant prognostic marker, and is a potential diagnostic marker for glutamine-dependent NSCLC [209]. However, toxicity in healthy cells slowed progress in bringing SLC1A5 inhibitors into clinical use [211].

Finally, targeting glutamine metabolism is also proposed to be effective in eradicating the CSC populations thought to be responsible for cancer relapse. Glutamine deprivation was found to diminish the proportion of CSC-like cells in various cancers including NSCLC, pancreatic cancer, and GBM [212]. Pharmacological inhibition of GLS was also shown to be effective in eradicating the GBM stem-like cell population, which is thought to be responsible for therapy resistance and tumour recurrence [213].

## 7. Targeting Stromal Components

### 7.1. CAFs

In the ‘reverse Warburg’ effect, CAFs undergo aerobic glycolysis and release lactate to fuel tumour OXPHOS [73]. Since this involves lactate export via MCT-4 into the TME, and subsequent lactate import via MCT-1 into the tumour, this metabolic crosstalk can potentially be blocked by MCT-1/MCT-4 inhibition. MCT-4 inhibition is able to block lactate export from tumours, resulting in lactate build-up, intracellular acidification, and end-product inhibition of glycolytic enzymes, which cripples tumour metabolism [214]. Syrosingopine is a dual MCT-1/MCT-4 inhibitor that leads to lactate accumulation and LDH inhibition in a mouse model of liver cancer, leading to reduced NAD^+^ levels [214]. Most other small-molecule MCT inhibitors developed to date are specific to MCT-1, with one drug (AZD3965) currently in clinical trials. However, AZD3965 is ineffective when MCT-4 is expressed [215], thus restricting its application to tumours that lack MCT-4.

The glutamine-based ‘tumour-feeding’ by CAFs may also be targeted. Inhibition of GLUL in CAFs, together with GLS inhibition in cancer cells, led to a synergistic effect in reducing tumour weight and metastasis in ovarian cancer mouse models when compared to monotherapy by disrupting the metabolic crosstalk between CAFs and ovarian tumour cells [79]. Similarly, GLS inhibition (by CB-839 and BTPES) in TNBC showed high efficacy initially, as TNBC is greatly reliant on exogenous glutamine for metabolism [200]. However, resistant cells soon emerged, which were able to respond to decreased glutamine inhibition by uptake of extracellular, CAF-derived pyruvate to replenish the TCA cycle, thus nullifying the effects of GLS inhibition [216].

### 7.2. ECs

As discussed in earlier sections, aberrant vessel sprouting is contributed by altered CCM leading to changes in the TME that impairs the normal vascularisation process in ECs. Thus, targeting CCM is a strategy to achieve tumour vessel normalisation as a potential anti-cancer treatment. Since tumour ECs (TECs) are highly glycolytic compared to normal proliferating ECs [217], one strategy is a blockade of glycolysis by inhibiting the glycolytic activator PFKFB3 with 3PO. Treatment with 3PO was found to promote tumour vessel normalisation, impairing cancer metastasis by tightening the EC barrier, and enhanced the efficiency of chemotherapy [84].

The hyperproliferative nature of TECs is also proposed to lead to greater dependency on mitochondrial metabolism, especially given the intense competition for nutrients in the TME, where glycolysis is likely already maximised by relying on mitochondrial ATP production to sustain angiogenesis [218]. As such, treating proliferating ECs with the weak mitochondrial uncoupler, Embelin, led to reduced OXPHOS, impairing tumour growth, and decreased microvessel density in mouse tumour models [219].

## 8. Targeting Metabolic Flexibility as a Mechanism of Resistance to CCM Inhibitors

Unfortunately, targeting inherent metabolic dependencies in isolation has met with halting failures due to metabolic plasticity in cancer cells. Upon inhibition of a particular pathway, tumours may simply reprogram their metabolism and upregulate a separate compensatory pathway, allowing escape from dependency on a single pathway. For instance, when various cancer cells were exposed to a continuous glycolytic block with 2-DG, a recurrent reprogramming mechanism was observed that led to escape from glycolytic addiction and, hence, escape from 2-DG susceptibility. This involved sustained mTORC1 signalling, directing glucose flux via the PPP back into glycolysis, nullifying the glycolytic block via anaplerosis [220]. This heightened metabolic flexibility is promoted by the pre-existing genetic and epigenetic instability in cancer cells, leading to rapid metabolic adaptation in response to inhibitors, and, in turn, therapy resistance [221,222]. Evidently, this flexibility and ease of switching to other pathways to fuel their metabolism poses a great challenge in developing successful CCM-targeting strategies.

Uncovering these mechanisms and metabolic tendencies has revealed evolutionary canalisations that can be exploited using a ‘synthetic lethality’ approach. In this section, combinatorial strategies involving dual metabolic inhibition and metabolic inhibitors in combination with targeted therapy, chemotherapy, and immunotherapy will be highlighted with multiple combinations showing promising advancements in clinical trials (Table 1, Table 2, Table 3 and Table 4).

### 8.1. Dual Metabolic Pathway Inhibition

Dual inhibition of OXPHOS and glycolysis is able to effectively disrupt energy metabolism and has proven to be effective against tumour growth in multiple preclinical cancer models (Table 1). For example, dual inhibition of glycolysis with 2-DG and OXPHOS with metformin-inhibited tumour growth preclinically in a broad spectrum of tumour models, including breast cancer, prostate cancer, GBM, and sarcoma [223,224,225,226,227,228,229]. Similarly, HK2 depletion in HCC sensitises cells to metformin [230]. Moreover, treatment of OXPHOS-competent hereditary leiomyomatosis RCC with the complex I inhibitor IACS-010759 plus simultaneous inhibition of the glycolytic enzyme phosphogluconate dehydrogenase (PGD) led to synthetic lethality [231]. Likewise in CLL cells, simultaneous inhibition of OXPHOS by IACS-010759 and glycolysis by 2-DG had a more pronounced effect than either inhibitor alone, providing a strong biological rationale for dual metabolic inhibition to deprive cancer cells of ATP [185].

Apart from OXPHOS and glycolysis inhibitor combinations, other dual metabolic inhibitor combinations have also been investigated. Many tumours rely on glutamine-mediated TCA cycle anaplerosis as an alternative source of carbon, and targeting glucose metabolism alone leads to compensatory dependency on glutaminolysis, providing the biological rationale for dual glutaminolysis plus glycolysis blockade [273]. Simultaneous glutaminolysis blockade with CB-839 and glycolysis inhibition with the HK2 inhibitor 3-BP was investigated in mice with renal tumours. The data showed promising results, as dual inhibition was able to significantly reduce overall lesions, while neither drug alone did [233]. Another combination strategy is to couple OXPHOS inhibition together with metabolite transporter blockade. Dual inhibition of the lactate transporters MCT-1 and MCT-4 with syrosingopine was found to be synthetic lethal with metformin due to NAD^+^ depletion in a mouse model of liver cancer [214]. A phase I study of metformin plus GLUT4 inhibition with ritonavir for relapsed/refractory MM or CLL is due to be completed in October 2020 (NCT02948283).

### 8.2. Metabolic Inhibition and Cell Signalling Pathway Inhibition

As a common theme, oncogene-addicted tumours that become resistant to primary TKI therapy develop OXPHOS dependency [274], and OXPHOS inhibition has been shown to resensitise cells to TKI therapy [5,232,235]. As a result, BRAF inhibitor-resistant melanoma was susceptible to complex I inhibition with phenformin, which re-sensitised cells to BRAF inhibition [5,234]. Similarly, the cKIT inhibitor imatinib was synergistic with the mitochondrial inhibitor VLX600 to limit the growth of GIST in mouse models [235]. Promising preclinical data have led to clinical trials testing OXPHOS inhibitor and TKI inhibitor combinations.

An ongoing phase I trial is evaluating the safety and efficacy of phenformin in combination with dabrafenib and trametinib in *BRAF*-mutant melanoma (NCT03026517). Various trials with metformin plus *BRAF* inhibitor combinations are also ongoing for numerous tumour types, including melanoma (NCT01638676, NCT02143050). A randomised phase II trial was conducted for the use of metformin in combination with front-line EGFR-TKI for *EGFR*-mutant NSCLC, with promising results (NCT03071705). The median progression-free survival and overall survival were significantly longer in the EGFR-TKIs plus the metformin group compared to EGFR-TKIs group [236]. Similarly, an ongoing clinical trial is evaluating the safety and efficacy of OPB compounds in combination with targeted therapies in oncogene-addicted tumours (NCT03158324).

While some tumours upregulate OXPHOS, others are reported to increase glycolysis in response to primary TKI therapy [275]. As a result, studies using glycolysis inhibition plus TKI therapies have shown promising preclinical results (Table 2). PFKFB3 inhibition with 3PO effectively suppressed tumour growth in combination with multi-kinase inhibitors nintedanib and sunitinib in a mouse model of breast cancer [237]. Similarly, promising data has been shown for derivatives of 3PO, PFK158, and PFK15 in combination with vemurafenib for *BRAF*-mutant melanoma cells in-vitro [238], and in combination with rapamycin for AML, respectively [239], with plans for phase I/II trials of PFK158 in combination with targeted agents underway. Preclinical success had also been seen with glycolytic inhibition using 2-DG in combination with TKI therapy. For instance, 2-DG treatment was synergistic with and able to resensitise NSCLC to the second-generation irreversible EGFR-TKI afatinib [240]. 2-DG was also synergistic with sorafenib in inducing apoptosis of sorafenib-resistant HCC cells [241]. Similarly, HK2 silencing in combination with sorafenib produced the same effect in inhibiting HCC tumour growth [230].

Glutamine metabolism also plays a key role in metabolic reprogramming in treatment resistance to targeted therapy with raised GSH levels conferring tumours with greater ability to maintain redox homeostasis [59,275]. Furthermore, another proposed mechanism by which oncogene-addicted cancers develop resistance to TKI therapy is via switching to glutaminolysis as a means of increasing OXPHOS [50]. This is shown by the development of OXPHOS dependency with acquired resistance to BRAF inhibitors upon treatment in *BRAF*-mutant melanoma [50]. Interestingly, this shift toward oxidative metabolism is associated with a switch from glucose to glutamine metabolism, suggesting a possible link between these two metabolic pathways and resistance to targeted therapies. Treatment of melanoma cells with the GLS inhibitor BPTES enhanced the anti-tumour activity of BRAF inhibition by suppressing the switch toward glutamine metabolism and OXPHOS [50] (Table 2).

Building on this idea, ongoing phase I/II studies of the GLS inhibitor CB-839 in combination with osimertinib is being tested for *EGFR-mutant* NSCLC (NCT03831932) and CB-839 with palbociclib for solid tumours (NCT03965845) (Table 2). In RCC, preclinical models showed CB-839 synergism with cabozantinib (Cabo), a VEGFR2/MET/AXL inhibitor, leading to inhibition of metabolic pathways and enhanced anti-tumour activity [242,243]. A completed phase I study reported encouraging clinical activity and tolerability in heavily pre-treated RCC patients, comparing favourably to historical Cabo monotherapy (NCT02071862) [243]. This has led development into a randomised phase II study of Cabo with CB-839/placebo in RCC (CANTATA, NCT0342821).

Preclinical studies showed that treatment with mTOR kinase inhibitors led to acquired resistance associated with compensatory upregulation of glutamine metabolism. Combined inhibition of mTOR kinase and GLS resulted in synergistic tumour cell death and growth inhibition in mice bearing GBM [244] (Table 2). This concept is also utilised in clinical trials for treating RCC. A prior phase Ib study of the GLS inhibitor CB-839 plus the mTOR inhibitor everolimus demonstrated impressive disease control rates, prompting a further randomised phase II study evaluating the efficacy of everolimus with CB-839/placebo in RCC (ENTRATA (CB-839 with Everolimus vs. Placebo with Everolimus in Patients With RCC), NCT03163667). Preliminary data show promising results with tolerable safety profiles in heavily-treated patients, including those refractory to multiple TKIs and immune checkpoint inhibitors.

mTOR inhibition may also be combined with biguanides such as metformin. While Akt activity is compensatorily induced by mTOR inhibition, metformin is able to counteract this upregulation of Akt by activating AMPK, providing scientific rationale for combining these two classes of agents [249,276]. Several preclinical studies combining mTOR inhibition with mitochondrial inhibitors, such as metformin, showed synergistic inhibition of tumour growth in pancreatic and breast cancer [245,246,247]. However, a completed phase Ib trial showed that the combination of everolimus and metformin is poorly tolerated in patients with advanced cancer [248]. On the other hand, the combination of metformin with other mTOR inhibitors have been more promising. Two completed phase I studies investigating the combination of metformin plus temsirolimus for advanced/refractory cancers (NCT01529593) and solid tumours or lymphoma (NCT00659568) showed the combination was well-tolerated with modestly promising effectiveness [249] (Table 2). Plans for phase II trials for this combination is underway. Finally, phase I trials for treating solid tumours with metformin in combination with sapanisertib, another mTOR1/2 inhibitor, is now recruiting (NCT03017833) (Table 2).

### 8.3. Metabolic Inhibition Plus Chemotherapy

Metabolic inhibitors are thought to reduce cancer cells therapy resistance by reducing the levels of key metabolites necessary for DNA damage repair, thus enhancing chemotherapy sensitivity [7,277]. This highlights the rationale for metabolic inhibitor plus chemotherapy combinations to increase the efficacy of chemotherapy with an array of combinations in the developmental pipeline (Table 3).

For example, HK2 inhibitor, 3-BP, induces the imbalance of intracellular redox through glycolytic inhibition, leading to large amounts of ROS production and intracellular accumulation [278]. 3-BP was found to be a chemosensitizer in combination with cisplatin and oxaliplatin in cell models of CRC [250] (Table 3). Similarly, 3-BP is able to act as a chemosensitizer in combination with the first-line chemotherapy drug for CRC, 5-fluorouracil [251]. Another proposed mechanism of 3-BP activity is via decreasing ATP production, leading to reduced activity of the ABC transporters, which are ATP-dependent efflux pumps, thereby restoring sensitivity to chemotherapy drugs such as daunorubicin, mitoxantrone, and doxorubicin in various cancer cell lines [253]. Although 3-BP usage in human studies was halted due to toxicities, this illustrates a proof-of-concept in the synergistic effect of glycolytic inhibition with chemotherapy.

Glycolysis inhibitor, 2-DG, was found to synergise with etoposide-induced cytotoxicity in the treatment of mouse models of lymphoma [254]. Similarly, combined treatment with 2-DG and doxorubicin enhanced the in-vitro efficacy of breast cancer radiotherapy [255]. Early studies also demonstrated synergism in treating mouse xenografts of human osteosarcoma and lung cancer cell lines with 2-DG + doxorubicin and 2DG + paclitaxel, respectively [256], leading to a phase I trial of 2-DG in combination with docetaxel was studied in various malignancies, which demonstrated safety and feasibility [257]. Finally, a novel candidate for glycolytic inhibition is via PKM2 modulation with early preclinical studies demonstrating synergism with cisplatin in overcoming chemoresistance in cervical [258] and bladder cancer [259] as well as in combination with docetaxel in human lung cancer xenografts in mice [260].

Targeting glutaminolysis to counter GSH production is also a promising strategy to re-sensitise tumours to chemotherapy [59]. Particular cancer types, such as TNBC, are also associated with elevated GLS expression and glutamine dependency [279,280]. CB-839 synergises with paclitaxel by reversing GLS-dependent mechanisms that lead to taxane resistance [200]. Leveraging on this, a phase I showed the paclitaxel + CB-839 combination was well-tolerated and demonstrated clinical activity in heavily pre-treated TNBC patients [261]. This led to a further phase II study of Paclitaxel + CB-839 in advanced TNBC (NCT03057600) with preliminary findings, demonstrating clinical activity and tolerability [262]. Other GLS inhibition plus chemotherapy combinations are also being investigated, reaching various stages of clinical trials (Table 3).

Another metabolic inhibitor plus chemotherapy combination that shows promising data is completed via TCA cycle inhibition. CPI-613 (Devimistat, Rafael Pharmaceuticals, Cranbury, NJ, USA) is a novel lipoic acid analogue that inhibits pyruvate dehydrogenase (PDH) and α-KG dehydrogenase enzymatic complexes. Favourable clinical trial data led to FDA granting CPI-613 orphan drug designation in pancreatic cancer, AML, myelodysplastic syndrome (MDS), peripheral T-cell lymphoma, and Burkitt’s lymphoma, and CPI-613 in combination with various chemotherapy regimens have been tested in clinical trials for PDAC and AML. Currently, a pivotal multicentre, open-label, randomised phase III trial (AVENGER 500, NCT03504423) is ongoing to evaluate the efficacy and safety of CPI-613 in combination with a modified FOLFIRINOX regimen for the first-line treatment of patients with metastatic pancreatic cancer [263].

A separate phase III trial is also ongoing to study CPI-613 in patients with AML (ARMADA 2000, NCT03504410). The ARMADA trial follows favourable results from multiple phase I/II studies of CPI-613 and high dose cytarabine and mitoxantrone given to relapsed or refractory AML patients. Given the favourable safety profile of this combination and the promising response achieved in these trials, further clinical evaluation is warranted.

Various combinations of metformin plus chemotherapy regimens have also shown favourable effects in preclinical models with some combinations reaching clinical trials [57,265]. The combination of metformin plus 5-fluorouracil for treating refractory CRC had completed a phase II trial (NCT01941953) [266]. Metformin in combination with the topoisomerase I inhibitor irinotecan is also currently being evaluated in a phase II trial for refractory CRC (NCT01930864). Metformin plus neo-adjuvant systemic therapy in HER2 positive breast cancer is being studied in a randomised phase II trial (NCT03238495, HERMET trial). Another randomized, open-label phase II trial investigating the role of metformin in addition to enzalutamide for castration-resistant prostate cancer is currently recruiting (IMPROVE trial, NCT02640534). OXPHOS inhibition with OPB-111077 is currently underway in a phase I trial for diffuse large B cell lymphoma (DLBCL) in combination with bendamustine and rituximab (NCT04049825).

### 8.4. Metabolic Inhibition Plus Immunotherapy

Anti-tumour activity of immune checkpoint inhibition may be enhanced by metabolic modulation of the TME. Promising preclinical data combining CB-839 with ICB has led to an ongoing phase I/II study of CB-839 in combination with nivolumab in immunogenic tumours including melanoma, RCC, and NSCLC (NCT02771626) (Table 4) [270]. Preliminary analyses demonstrated that CB-839 was well-tolerated when combined with nivolumab. Of note, of the eight evaluable RCC patients, 75% achieved SD on combination therapy, all of whom were progressing on a checkpoint inhibitor at study entry. Another area of focus is NSCLC containing activating mutations in the NRF2/KEAP1 pathway. A key subset of these activated genes upregulate glutamine metabolism and dependency. This clear mechanistic rationale had spurred a randomised double-blind phase II study evaluating CB-839 in combination with standard of care chemo-immunotherapy in this subgroup of NSCLC patients (NCT04265534).

Amino acid metabolism can also be targeted in combination with immunotherapy. Arg1 upregulation in MDSCs depletes T cells of l-arginine in the tumour milieu [118], limiting T cell anti-tumour activity. CB-1158 is an Arg1 inhibitor that reversed myeloid-mediated T cell inhibition in-vitro and suppressed tumour growth in-vivo. Furthermore, CB-1158 treatment was synergistic in combination with checkpoint blockade in multiple mouse models of cancer, resulting in increased tumour-infiltrating CD8^+^ T cells and NK cells, inflammatory cytokines, and expression of IFN-inducible genes [271]. This has led on to a phase I/II trial of CB-1158 in combination with pembrolizumab in solid tumours (NCT02903914). Currently, CB-1158 has been well-tolerated and achieves on-target inhibition, resulting in increases in plasma arginine [281]. Another phase I/II trial investigating CB-1158 plus subcutaneous daratumumab, compared to daratumumab monotherapy, in relapse or refractory MM, is currently recruiting (NCT03837509) (Table 4).

Another enzyme, indoleamine 2,3-dioxygenase (IDO), catalyses the rate-limiting step in tryptophan oxidation. This generates kynurenine, which simultaneously depletes T cells of the amino acid while promoting exerting immunosuppression and induces T cell apoptosis [93,106]. Furthermore, kynurenine acts as a ligand for the aryl hydrocarbon receptor (AHR), leading to Treg differentiation [282] and pro-tumourigenic effects. IDO inhibitors seek to prevent tryptophan depletion to reduce the production of immunosuppressive kynurenine. Multiple phase I/II trials showed encouraging results with small molecule inhibitors of IDO1, such as epacadostat, with improved responses to anti-PD-1 therapy (Table 4). However, recent results from ECHO-301, the first large phase III trial to evaluate the efficacy of epacadostat in combination with pembrolizumab in advanced melanoma, showed no indication that epacadostat provided an additional benefit. Thus, the current usefulness of IDO1 inhibition to enhance anti-PD-1 therapy remains uncertain. Other IDO1 inhibitors are being developed and, in earlier phase trials, including navoximod, currently being tested in a phase Ib trial for solid tumours in combination with atezolizumab [272] (Table 4).

Another immunometabolism-targeting pathway is via A2AR. CPI-444 (ciforadenant) is an A2AR antagonist that can inhibit immuno-suppressive effects of adenosine on T cells, NK cells, macrophages, and DCs. CPI-444 is currently in early-phase clinical trials in combination with atezolizumab for advanced RCC, prostate cancer, and NSCLC (NCT02655822, NCT03337698) and in combination with daratumumab for relapsed or refractory MM (NCT04280328).

## 9. Conclusions

Nearly a decade ago, cancer therapy entered a new era with the discovery of two additional hallmarks of cancer, ‘reprogramming energy metabolism’ and ‘evading immune response.’ Following intensive research and several ground-breaking discoveries, checkpoint inhibitor therapy has emerged as frontline therapy in multiple tumour types [283]. Targeting altered cell metabolism is also recognized as a potential means of achieving therapeutic selectivity due to the fundamental metabolic differences between normal and cancer cells. Yet, therapeutic advances have been much more modest on the metabolic front partly due to the nuances in deciphering complex and interconnected metabolic pathways. Moreover, the dynamic metabolic crosstalk between cancer cells and the TME, including the immune system, adds further layers of complexity to metabolic inhibition.

Nevertheless, important strides have been made toward the clinical application of metabolic inhibitors, which can be credited to the tremendous ongoing effort by researchers to tease out the intricacies of tumour metabolism and identify compounds with favourable pharmacokinetic and safety profiles. For one, the field has evolved from a small selection of agents with very narrow therapeutic windows, such as arsenic trioxide and 2-DG, to a wide selection of candidates targeting various pathways in which many have demonstrated tolerability in clinical trials. Glutamine has been identified as an excellent therapeutic target due to its contributions to both OXPHOS and glycolysis, and, not unexpectedly, the potent and selective GLS inhibitor, CB-839, has demonstrated great potential in the developmental pipeline. It has now entered phase II studies, and the strategic direction, which has been selected for its future development is in combination with chemotherapy, targeted therapies, and immunotherapy [202,243,261].

However, the therapeutic success of cell metabolism inhibitors cannot be solely reliant on the discovery of compounds with favourable pharmacologic properties. Biomarkers help stratify tumours, according to their metabolic dependencies and are a crucial element of patient selection for metabolic inhibition. To date, p53 or LKB1 loss, homozygous deletion of *ENO1* and *SMARCA4* mutations are strong candidate biomarkers of OXPHOS inhibition [186,187,188]. On the other hand, despite the excellent progress in the development of glutaminolysis inhibitors, a companion biomarker has yet to be established [284].

The discovery of altered CCM as a mechanism of resistance to standard anti-cancer therapies is arguably the most significant advancement in the field, serving as a platform for the discovery of rational synthetic lethality combinations to overcome therapeutic resistance to conventional chemotherapy and targeted therapies. Apart from targeting mechanisms of secondary resistance, preclinical studies have raised the possibility of eradicating notoriously therapy-resistant CSCs [163,285]. Several of these combinations have demonstrated both safety as well as promising activity in clinical trials [267]. Due to the synergistic action of both compounds, synthetic lethality strategies may permit the use of lower drug doses compared to single agent blockade, potentially mitigating drug toxicities.

Cancer cells have the innate ability to escape the inhibition of a particular metabolic pathway by upregulating compensatory pathways or deriving alternative routes of nutrient supply. At present, the prevailing plasticity of the metabolic circuitry is thought to be the greatest limiting factor in the success of metabolic inhibition, rendering the singular targeting of metabolic pathways ineffective. Dual metabolic inhibition is a promising strategy to overcome this, especially combinations involving the inhibition of glycolysis and OXPHOS, or glycolysis and glutaminolysis [185,223,230,231,233]. Though supported by robust preclinical data, the clinical evaluation of dual metabolic inhibitor strategies remain in their infancy stages [286].

Emerging data have led to observations that metabolic plasticity is not merely limited to cancer cells, but also involves the surrounding TME, especially the metabolically versatile immune milieu. Altered CCM creates an unfavourable microenvironment, resulting in immunosuppression and polarisation toward pro-tumourigenic cell types. Competing energetic requirements of tumour cells and the immune environment result in the limited availability of key nutrients, blocking T cell activation and proliferation. Furthermore, the immune checkpoint proteins on tumour cells, PD-1 and CTLA-4, suppress T cell metabolism. Hence, it is now evident that specific metabolic dependencies of immune cells lead to cancer immune evasion, which is a crucial discovery that may be exploited in order to enhance anti-cancer immune responses. This validates treatment strategies that are underway to evaluate various combinations of immune checkpoint blockades and metabolic inhibitors [79,270,272].

Although the development of metabolic inhibitors has been fraught with challenges and disappointments, significant momentum has been gained in recent times, all the more so with the discovery of effective drug combinations. With the aid of further translational studies and well-designed therapeutic strategies, the routine use of metabolic inhibitors in the clinic may become a reality in the near future.

## Figures and Tables

**Figure 1 molecules-25-04831-f001:**
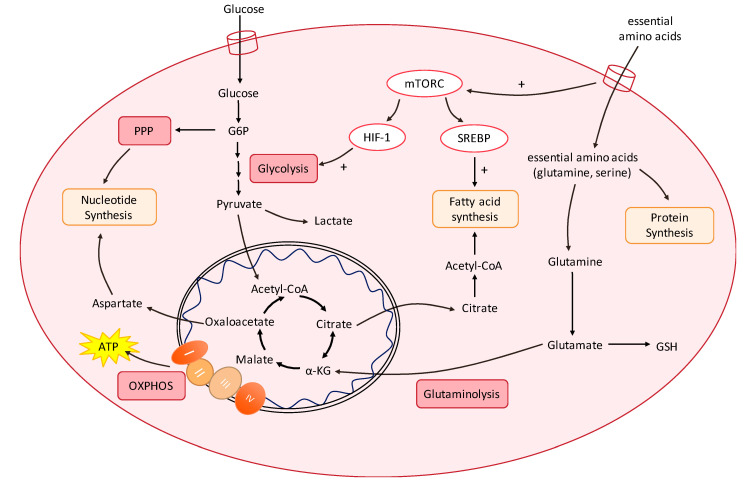
Overview of cancer cell metabolic reprogramming. Cancer cells require extensive metabolic reprogramming to fuel anabolic growth via increased nucleotide biosynthesis, protein synthesis, and FA synthesis. There is elevated glycolysis even under aerobic conditions (Warburg effect), which allows for the production of intermediates to be channelled into the PPP for nucleotide biosynthesis. However, a majority of tumours still retain oxidative capacity to produce ATP via OXPHOS. Glutaminolysis is also upregulated in many tumours for the production of α-KG to fuel the TCA cycle. Increased glutaminolysis also produces glutathione (GSH) to defend against oxidative stress. Central to these metabolic changes is the PI3K/Akt/mTOR pathway. Downstream effectors that are activated by mTORC signalling include the transcription factors HIF-1 and SREBP.

**Figure 2 molecules-25-04831-f002:**
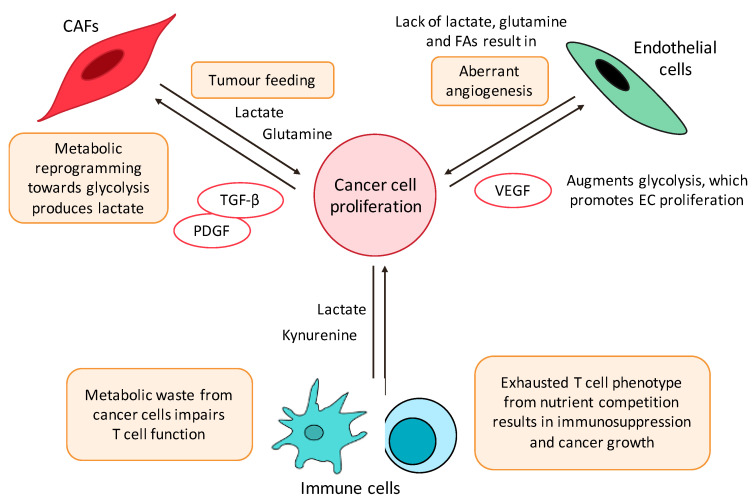
Key players of the metabolic crosstalk in the TME. Key players involved in the extensive, bidirectional crosstalk between tumour cells and the TME include CAFs, ECs, and immune cells. Tumours release factors such as PDGF and TGF-β, causing metabolic reprogramming in CAFs towards aerobic glycolysis, releasing energetic substrates such as lactate into the TME in a phenomenon known as ‘tumour-feeding.’ Meanwhile, tumour depletion of lactate, glutamine, and FAs in the TME lead to EC aberrant angiogenesis, which promotes proliferation and metastasis. VEGF is also released by tumours to promote EC proliferation. Tumour cells also induce metabolic changes to immune cells and cause immunosuppression. This is due to metabolic competition between immune cells and tumours for the same nutrients, producing an ‘exhausted’ T cell phenotype. Metabolic wastes, including lactate and kynurenine, are also released and impair T cell function, causing polarisation towards pro-tumorigenic T cell subtypes.

**Figure 3 molecules-25-04831-f003:**
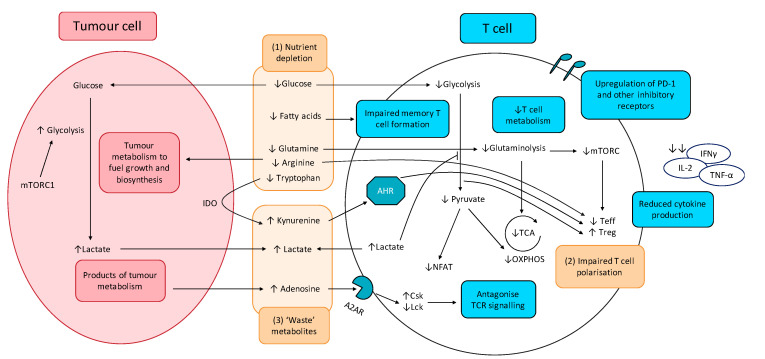
Effect of tumour metabolism on T cell function. (1) Altered cancer cell metabolism results in nutrient competition, depriving T cells of essential nutrients essential for robust anti-tumour activity, including glucose and key amino acids. Resultant exhausted T cell phenotype shows upregulation of inhibitory receptors including PD-1, CTLA-4, TIM-3, LAG-3, and TIGIT, impaired production and release of effector cytokines (IFNγ, IL-2, and TNF-α), as well as impaired degranulation. (2) Depletion of key nutrients and aberrant metabolite signalling promotes pro-tumourigenic T cell phenotypes. (3) Cancer cell metabolism produces lactate and other ‘waste’ metabolites that inhibit T cell function and promote T cell exhaustion.

**Table 1 molecules-25-04831-t001:** Dual metabolic inhibitor combinations for cancer therapy.

Targeted Metabolism	Metabolic Inhibitor 1	Metabolic Inhibitor 2	Preclinical Data	Clinical Data
OXPHOS + Glycolysis	Metformin	2-deoxyglucose (2-DG)	Breast, prostate, GBM, sarcoma, PDAC, oesophageal, ovarian cancers [223,224,225,226,227,228,229]	
		HK2 deletion	HCC [230]	
	IACS-010759 (complex I inhibitor)	Phosphogluconate dehydrogenase (PGD) inhibition	Hereditary leiomyomastosis RCC [231]	
		2-DG	CLL [185]	
	BAY87-2243 (B87)	Dimethyl α-KG (DMKG)	Multiple: NSCLC, CRC, glioma, breast, sarcoma [232]	
Glutaminolysis + Glycolysis	CB-839	3-BP (HK2 inhibitor)	Renal [233]	
OXPHOS + Metabolite Transporter	Metformin	Syrosingopine (MCT-1 and MCT-4 inhibitor)	Liver [214]	
		Ritonavir (GLUT4 inhibition)		Phase I—MM, CLL (NCT02948283)

**Table 2 molecules-25-04831-t002:** Metabolic inhibitors in combination with targeted therapy for cancer therapy.

Targeted Metabolism	Metabolic Inhibitor	Cell Signalling Pathway Inhibitor	Preclinical Data	Clinical Data
OXPHOS	Phenformin	BRAF inhibition (Dabrafenib + Trametinib)	Melanoma [5,234]	Phase I—Melanoma (NCT03026517)
	VLX600 (mitochondrial inhibitor)	cKIT inhibition (Imatinib)	GIST [235]	
	IACS-010759 (complex I inhibitor)	Ibrutinib	MCL [48]	
	Metformin	BRAF TKI (vemurafenib or dabrafenib + trametinib)		Phase I/II—Melanoma (NCT01638676) Phase I/II—Melanoma (NCT02143050)
		EGFR TKI (erlotinib, afatinib or gefitinib)		Phase II—NSCLC (NCT03071705) [236]
	OPB compounds (OPB-51602, OPB-1110077)	EGFR TKI Cell signalling pathway inhibitors		Phase I—NSCLC (NCT01184807) [193] Phase IIa—Oncogene-addicted cancers (NCT03158324)
Glycolysis	3PO	Nintedanib, sunitinib	Breast [237]	
	PFK158 (PFKFB3 inhibitor)	Vemurafenib	Melanoma [238]	
	PFK15 (PFKFB3 inhibitor)	Rapamycin	AML [239]	
	2-DG	Afatinib	NSCLC [240]	
	2-DG HK2 silencing	Sorafenib	HCC [230,241]	
Glutaminolysis	GLS inhibition (BPTES, CB-839)	BRAF TKI	Melanoma [50]	
		Osimertinib		Phase I/II—NSCLC (NCT03831932)
		Erlotinib		Phase I—NSCLC (NCT02071862)
		Palbociclib		Phase I/II—Solid tumours (NCT03965845)
		Cabozantinib	Multiple: melanoma, glioma, NSCLC, sarcoma, PDAC, prostate [242]	Phase I—RCC (NCT02071862) [243] Phase II—RCC (CANTATA: NCT03428217)
Metabolic inhibition + mTOR Pathway inhibition	Compound 968	Rapamycin	GBM [244]	
	CB-839	Everolimus		Phase Ib—RCC (NCT02071862) Phase II—RCC (ENTRATA: NCT03163667)
	Metformin	Rapamycin	Pancreatic [245]	
		Everolimus	Breast [246,247]	Phase Ib—Solid tumours [248]
		Temsirolimus		Phase I—advanced/refractory cancers (NCT01529593), solid tumours or lymphoma (NCT00659568) [249]
		Sapanisertib (TAK-228) mTOR1/2 inhibitor		Phase I—solid tumours (NCT03017833)

**Table 3 molecules-25-04831-t003:** Metabolic inhibitors in combination with chemotherapy.

Targeted Metabolism	Metabolic Inhibitor	Chemotherapy	Preclinical Data	Clinical Data
Glycolysis	3-BP (HK2 inhibitor)	Platinum drugs (cisplatin, oxaliplatin)	CRC [250]	
		5-fluorouracil	CRC [251]	
		Doxorubicin	Neuroblastoma [252]	
		Daunorubicin, mitoxantrone, doxorubicin	MM, AML, HCC [253]	
	2-DG	Etoposide	Lymphoma [254]	
		Doxorubicin + radiotherapy	Breast [255]	
		Doxorubicin, paclitaxel Docetaxel	Osteosarcoma, NSCLC [256]	Phase I—various [257]
	PKM2 modulation	Cisplatin	Cervical [258] Bladder [259]	
		Docetaxel	Lung [260]	
Glutaminolysis	CB-839	Paclitaxel	TNBC [200]	Phase I—TNBC (NCT02071862) [261] Phase II—TNBC (NCT03057600) [262]
		Docetaxel		Phase I—NSCLC (NCT02071862)
		Cepecitabine		Phase I/II—solid tumours (NCT02861300)
Mitochondrial Metabolism	CPI-613/Devimistat (PDH and α-KG dehydrogenase complex inhibitor)	FOLFIRINOX (oxaliplain, folinic acid, irinotecan, fluorouracil)		Phase I—PDAC (NCT01835041) [263] Phase III—PDAC (AVENGER 500 trial, NCT03504423)
		Cytarabine + mitoxantrone		Phase III—AML (ARMADA 2000 trial, NCT03504410) [264]
	Metformin	Doxorubicin	Prostate, lung [265]	
		Carboplatin	NSCLC [57]	
		5-fluorouracil		Phase II—CRC (NCT01941953) [266]
		Irinotecan		Phase II—CRC (NCT01930864)
		Neo-adjuvant chemotherapy (TCH+P)		Phase II—HER2-positive breast (HERMET trial, NCT03238495)
		Radiotherapy		Phase II—prostate (NCT02945813)
		Enzalutamide		Phase II—prostate (IMPROVE trial, NCT02640534)
	IACS-010759 (complex I inhibitor)	Cytarabine + doxorubicin	AML [267]	
	OPB-111077	Bendamustine + rituximab		Phase I—diffuse large B cell lymphoma (DLBCL) (NCT04049825)
	Dicholoroacetate (PDK2 inhibitor)	Paclitaxel	NSCLC [268]	
		5-fluorouracil	CRC [269]	
Other enzymes	CB-1158/INCB001158 (Arg1 inhibitor)	Chemotherapy		Phase I/II—solid tumours (NCT03314935)
	Indoximod/1-methyl-d-tryptophan (IDO1 inhibitor)	Taxane chemotherapy		Phase II—breast (NCT01792050)
		Gemcitabine		Phase I/II—PDAC (NCT02077881)

**Table 4 molecules-25-04831-t004:** Metabolic inhibitors in combination with immunotherapy.

Targeted Metabolism	Metabolic Inhibitor	Immunotherapy	Preclinical Data	Clinical Data
Glutaminolysis	CB-839	Anti-PD-1, anti-PD-L1	Colon [270]	
		Nivolumab		Phase I/II—melanoma, RCC, NSCLC (NCT02771626)
		Pembrolizumab + carboplatin + pemetrexed		Phase II—NSCLC (NCT04265534)
	JHU083	Anti-PD-1	Lymphoma, colon, melanoma [104]	
Amino acid metabolism	CB-1158/INCB001158 (Arg1 inhibitor)	Anti-PD-1 Pembrolizumab Daratumumab	Solid tumours [271]	Phase I/II—solid tumours (NCT02903914) Phase I/II—MM (NCT03837509)
	Epacadostat/INCB024360 (IDO1 inhibitor)	Checkpoint inhibitors (various)		Phase I/II—solid tumours (multiple clinical trials)
		Pembrolizumab		Phase III—melanoma (NCT02752074) [272] Phase III—melanoma, urothelial carcinoma, HNSCC (Keynote-ECHO trials: NCT02752074, NCT03361865, NCT03374488, NCT03358472)
	Navoximod/GDC-0919 (IDO1 inhibitor)	Atezolizumab		Phase Ib—solid tumours (NCT02471846, NCT02048709)
Other	CPI-444/ciforadenant (A2AR antagonist)	Atezolizumab		Phase I—RCC, prostate (NCT02655822) Phase I/II—NSCLC (NCT03337698)
		Daratumumab		Phase I—MM (NCT04280328)

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
