# Peer review of "Targeting Metabolism in Cancer Cells and the Tumour Microenvironment for Cancer Therapy"

_molecules, 2020, doi:10.3390/molecules25204831_

Round 1

Reviewer 1 Report

In this manuscript, Li J et al. reviewed recent literature and progresses in the field of metabolism in cancer cells and TME. The topic is of high interest with increasing inputs from research labs all over the world. In general, this manuscript nicely summarized such progresses made in the past decade. Concerns are raised during this review for authors to improve.

Concerns:

  1. As the title implies, this review focuses on cancer cells and tumor microenvironment for cancer therapy. However, the same group just recently published a similar review article in Antioxidants & Redox Signaling on cell metabolism (Targeting Cell Metabolism as Cancer Therapy. Natalie Y L Ngoi, Jie Qing Eu, Jayshree Hirpara, Lingzhi Wang, Joline S J Lim, Soo-Chin Lee, Yaw-Chyn Lim, Shazib Pervaiz, Boon Cher Goh, Andrea L A Wong. Antioxid Redox Signal. 2020 Feb 10;32(5):285-308. PMID: 31841375). A significant portion of the information present in this manuscript on cancer cell metabolism is overlapping with the above referred published review article. The authors should either focus on TME or properly make clear the new info/addition to the current manuscript.
  2. There are 7 co-authors listed in this manuscript, comparing to a 10 co-author list in the referred published review articles in A&RS. If some middle authors contributed superficially, or with barely moderate contribution, then these co-authors should be moved to the Acknowledgements section.
  3. The Conclusion and Future Perspectives section is overly long.
  4. Table 1A-1D should be re-organized as Tables 1-4.

Author Response

On behalf of the authors, I would like to thank you for your helpful comments and suggestions. We have made the suggested revisions to the manuscript.

Responses to Reviewer 1

  1. As the title implies, this review focuses on cancer cells and tumor microenvironment for cancer therapy. However, the same group just recently published a similar review article in Antioxidants & Redox Signaling on cell metabolism (Targeting Cell Metabolism as Cancer Therapy. Natalie Y L Ngoi, Jie Qing Eu, Jayshree Hirpara, Lingzhi Wang, Joline S J Lim, Soo-Chin Lee, Yaw-Chyn Lim, Shazib Pervaiz, Boon Cher Goh, Andrea L A Wong. Antioxid Redox Signal. 2020 Feb 10;32(5):285-308. PMID: 31841375). A significant portion of the information present in this manuscript on cancer cell metabolism is overlapping with the above referred published review article. The authors should either focus on TME or properly make clear the new info/addition to the current manuscript.

In accordance to the reviewer’s suggestions, this current Review aims to adopt a much greater focus on the role of the stromal and immune components of the TME. In particular, section 4.3 is newly added (natural killer cells and neutrophils) under “4. Tumour Immune Microenvironment”.

Section 2.3.1 “Resistance to Cell Signalling Pathway Inhibitors” had also been revised, as per the reviewer’s suggestions. This section has been trimmed to reduce the overlap with the last review and we have focused instead on non-CSC mediated resistance mechanisms.

  1. There are 7 co-authors listed in this manuscript, comparing to a 10 co-author list in the referred published review articles in A&RS. If some middle authors contributed superficially, or with barely moderate contribution, then these co-authors should be moved to the Acknowledgements section.

Thank you for your comment regarding the author lists. As all authors have contributed to the content and review of the manuscript, we have decided to keep all 7 co-authors.

  1. The Conclusion and Future Perspectives section is overly long.

The Section “9. Conclusion and Future Perspective” has been trimmed as per the reviewer’s suggestions. Specifically, areas of overlap with previous review reduced as much as possible.

  1. Table 1A-1D should be re-organized as Tables 1-4.

Thank you for the suggestion. As advised, we have re-organised the Tables into 4 separate tables (Tables 1-4).

Reviewer 2 Report

This review article discusses metabolic reprogramming in cancer including the tumor microenvironment and how they can be targeted for cancer therapy.  It is quite comprehensive in scope, giving some examples for each of the topic.  It is well-written and the topics are very timely and interesting.  The references are also appropriate.  

Author Response

We thank the reviewer for the positive comments.

Reviewer 3 Report

The review "Targeting Metabolism in Cancer Cells and the Tumour Microenvironment for Cancer Therapy" by Jiaqi Li et al describes finely therapeutic opportunities and combinatorial therapeutic strategies related to dysregulated metabolism and metabolic crosstalk in cancer.

The review is well done and the conclusions are well driven from the data available.

Author Response

(The authors gave the same response as above.)

Reviewer 4 Report

This review by Jiaqi Li et al. provides an exhaustive up-to-date on cancer therapy targeting metabolism in cancer cells and tumor microenvironment. The first part of the paper describes the altered cancer cell metabolism and the crosstalk with the tumor microenvironment, highlighting the set of pathways activated in each case and its importance in tumor progression. The untangling of this metabolism is an opportunity to tackle cancer with different metabolic inhibitors. The second part of the paper focuses on this therapeutic opportunity and close with a concise conclusion and future perspectives. The different figures and tables presented are informative, well distributed in the article, and serve the purpose of helping the lector follow the explanation. The study is well-written, easy to follow, and educational. For this reason, I don't see any significant concern, recommending the publication of the manuscript in the present form.

Author Response

(The authors gave the same response as above.)

Round 2

Reviewer 1 Report

The authors have addressed raised concerns.